# MotifExplainer: A Motif-based Graph Neural Network Explainer

## Abstract

We consider the explanation problem of Graph Neural Networks (GNNs). Most existing GNN explanation methods identify the most important edges or nodes but fail to consider substructures, which are more important for graph data. One method considering subgraphs tries to search all possible subgraphs and identifies the most significant ones. However, the subgraphs identified may not be recurrent or statistically important for interpretation. This work proposes a novel method, named MotifExplainer, to explain GNNs by identifying important motifs, which are recurrent and statistically significant patterns in graphs. Our proposed motif-based methods can provide better human-understandable explanations than methods based on nodes, edges, and regular subgraphs. Given an instance graph and a pre-trained GNN model, our method first extracts motifs in the graph using domain-specific motif extraction rules. Then, a motif embedding is encoded by feeding motifs into the pre-trained GNN. Finally, we employ an attention-based method to identify the most influential motifs as explanations for the prediction results. The empirical studies on both synthetic and real-world datasets demonstrate the effectiveness of our method.

Graph neural networks (GNNs) have shown capability in solving various challenging tasks in graph fields, such as node classification, graph classification, and link prediction. Although many GNNs models Kipf & Welling (2016); Gao et al. (2018); Xu et al. (2018); Gao & Ji (2019); Liu et al. (2020) have achieved state-of-the-art performances in various tasks, they are still considered black boxes and lack sufficient knowledge to explain them. Inadequate interpretation of GNN decisions severely hinders the applicability of these models in critical decision-making contexts where both predictive performance and interpretability are critical. A good explainer allows us to debate GNN decisions and shows where algorithmic decisions may be biased or discriminated against. In addition, we can apply precise explanations to other scientific research like fragment generation. A fragment library is a key component in drug discovery, and accurate explanations may help its generation.

Several methods have been proposed to explain GNNs, divided into instance-level explainers and model-level explainers. Most existing instance-level explainers such as GNNExplainer Ying et al. (2019), PGExplainer Luo et al. (2020), Gem Lin et al. (2021), and ReFine Wang et al. (2021) produce an explanation to every graph instance. These methods explain pre-trained GNNs by identifying important edges or nodes but fail to consider substructures, which are more important for graph data. The only method that considers subgraphs is SubgraphX Yuan et al. (2021), which searches all possible subgraphs and identifies the most significant one. However, the subgraphs identified may not be recurrent or statistically important, which raises an issue on the application of the produced explanations. For example, fragment-based drug discovery (FBDD) Erlanson et al. (2004) has been proven to be powerful for developing potent small-molecule compounds. FBDD is based on fragment libraries, containing fragments or motifs identified as relevant to the target property by domain experts. Using a motif-based GNN explainer, we can directly identify relevant fragments or motifs that are ready to be used when generating drug-like lead compounds in FBDD.

In addition, searching and scoring all possible subgraphs is time-consuming and inefficient. We claim that using motifs, recurrent and statistically important subgraphs, to explain GNNs can provide a more intuitive explanation than methods based on nodes, edges, or subgraphs.

This work proposes a novel GNN explanation method named MotifExplainer, which can identify significant motifs to explain an instance graph. In particular, our method first extracts motifs from a

given graph using domain-specific motif extraction rules based on domain knowledge. Then, motif embeddings of extracted motifs are generated by feeding motifs into the target GNN model. After that, an attention model is employed to select relevant motifs based on attention weights. These selected motifs are used as an explanation for the target GNN model on the instance graph. To our knowledge, the proposed method represents the first attempt to apply the attention mechanism to explain the GNN from the motif-level perspective. We evaluate our method using both qualitative and quantitative experiments. The experiments show that our MotifExplainer can generate a better explanation than previous GNN explainers. Furthermore, the efficiency studies demonstrate the efficiency advantage of our methods in terms of a much shorter training and inference time.

## 1 PROBLEM FORMULATION

This section formulates the problem of explanations on graph neural networks. Let $G_i = \{V, E\} \in \mathcal{G} = \{G_1, G_2, ..., G_i, ..., G_N\}$ denotes a graph where $V = \{v_1, v_2, ..., v_i, ...v_n\}$ is the node set of the graph and $E$ is the edge set. $G_i$ is associated with a $d$-dimensional set of node features $\boldsymbol{X} = \{\boldsymbol{x}_1, \boldsymbol{x}_2, ..., \boldsymbol{x}_i, ..., \boldsymbol{x}_n\}$, where $\boldsymbol{x}_i \in \mathbb{R}^d$ is the feature vector of node $v_i$. Without loss of generality, we consider the problem of explaining a GNN-based downstream classification task. For a node classification task, we associate each node $v_i$ of a graph $G$ with a label $y_i$, where $y_i \in Y = \{l_1, ..., l_c\}$ and $c$ is the number of classes. For a graph classification task, each graph $G_i$ is assigned a corresponding label.

### 1.1 BACKGROUND ON GRAPH NEURAL NETWORKS

Most Graph Neural Networks (GNNs) follow a neighborhood aggregation learning scheme. In a layer $\ell$, GNNs contain three steps. First, a GNN first calculates the messages that will be transferred between every node pair. A message for a node pair $(v_i, v_j)$ can be represented by a function $\theta(\cdot) : \boldsymbol{b}_{ij}^\ell = \theta(\boldsymbol{x}_i^{\ell-1}, \boldsymbol{x}_j^{\ell-1}, \boldsymbol{e}_{ij})$, where $\boldsymbol{e}_{ij}$ is the edge feature vector, $\boldsymbol{x}_i^{\ell-1}$ and $\boldsymbol{x}_j^{\ell-1}$ are the node features of $v_i$ and $v_j$ at the previous layer, respectively. Second, for each node $v_i$, GNN aggregates all messages from its neighborhood $\mathcal{N}_i$ using an aggregation function $\phi(\cdot) : \boldsymbol{B}_i^\ell = \phi\left(\{\boldsymbol{b}_{ij}^\ell | v_j \in \mathcal{N}_i\}\right)$. Finally, the GNN combine the aggregated message $\boldsymbol{B}_i^\ell$ with node $v_i$'s feature representation from previous layer $\boldsymbol{x}_i^{\ell-1}$, and use a non-linear activation function to obtain the representation for node $v_i$ at layer $l : \boldsymbol{x}_i^\ell = f(\boldsymbol{x}_i^{\ell-1}, \boldsymbol{B}_i^\ell)$. Formally, a $\ell$-th GNN layer can be represented by

$$\boldsymbol{x}_i^\ell = f\left(\boldsymbol{x}_i^{\ell-1}, \phi\left(\left\{\theta\left(\boldsymbol{x}_i^{l-1}, \boldsymbol{x}_j^{l-1}, \boldsymbol{e}_{ij}\right)\right\} \mid v_j \in \mathcal{N}_i\right\}\right)\right).$$

### 1.2 GRAPH NEURAL NETWORK EXPLANATIONS

In a GNN explanation task, we are given a pre-trained GNN model, which can be represented by $\Psi(\cdot)$ and its corresponding dataset $\mathcal{D}$. The task is to obtain an explanation model $\Phi(\cdot)$ that can provide a fast and accurate explanation for the given GNN model. Most existing GNN explanation approaches can be categorized into two branches: instance-level methods and model-level methods. Instance-level methods can provide an explanation for each input graph, while model-level methods are input-independent and analyze graph patterns without input data. Following previous works Luo et al. (2020); Yuan et al. (2021); Lin et al. (2021); Wang et al. (2021); Bajaj et al. (2021), we focus on instance-level methods with explanations using graph sub-structures. Also, our approach is model-agnostic. In particular, given an input graph, our explanation model can generate a subgraph that is the most important to the outcomes of a pre-trained GNN on any downstream graph-related task, such as graph classification tasks.

## 2 MOTIF-BASED GRAPH NEURAL NETWORK EXPLAINER

Most existing GNN explainers Ying et al. (2019); Luo et al. (2020) identify the most important nodes or edges. SubgraphX Yuan et al. (2021) is the first work that proposed a method to explain GNN models by generating the most significant subgraph for an input graph. However, the subgraphs identified by SubgraphX may not be recurrent or statistically important. This section proposes a novel GNN explanation method, named MotifExplainer, to explain GNN models based on motifs.

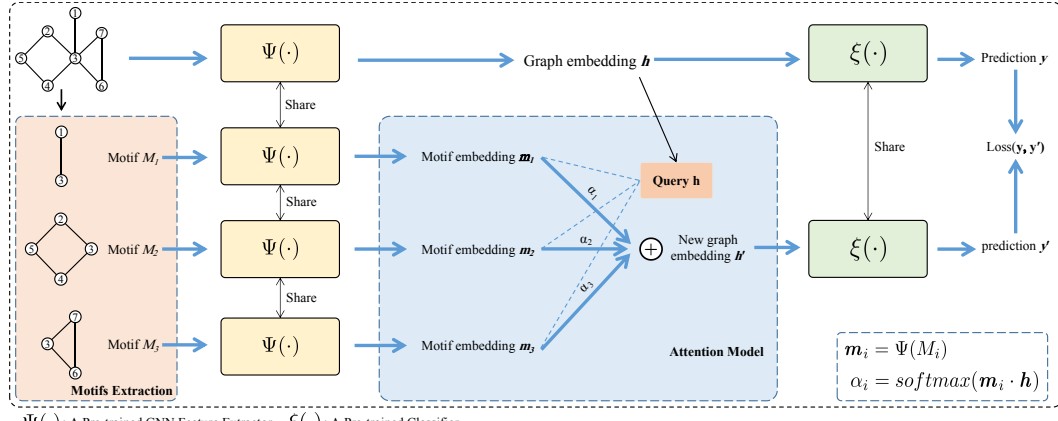

Figure 1: An illustration of the proposed MotifExplainer on graph classification tasks. Given a graph, we first extract motifs based on extraction rules. Then, motif embedding is generated for each motif by feeding it into the pre-trained GNN feature extractor. After that, we employ an attention layer that uses graph embedding as the query and motif embedding as keys and values, resulting in a new graph embedding. Finally, the loss is computed based on the new and the original predictions.

## 2.1 FROM SUBGRAPH TO MOTIF EXPLANATION

Unlike explanations on models for text and image tasks, a graph has non-grid topology structure information, which needs to be considered in an explanation model. Given an input graph and a trained GNN model, most existing GNN explainers, such as GNNExplainer Ying et al. (2019) and PGExplainer Luo et al. (2020), identify important edges and construct a subgraph containing all those edges as the explanation of the input graph. However, these models ignore the interactions between edges or nodes and implicitly measure the essence of substructures. SubgraphX Yuan et al. (2021) proposed to employ subgraphs for GNN explanation. It explicitly evaluates subgraphs and considers the interaction between different substructures. However, it does not use domain knowledge like motif information when generating the subgraphs.

A motif can be regarded as a simple subgraph of a complex graph, which repeatedly appears in graphs and is highly related to the function of the graph. Motifs have been extensively studied in many fields, like biochemistry, ecology, neurobiology, and engineering Milo et al. (2002); Shen-Orr et al. (2002); Alon (2007; 2019) and are proved to be important. A subgraph identified without considering domain knowledge can be ineffective for downstream tasks like fragment library generation in FBDD. Thus, it is desirable to introduce statistically important motif information to a more human-understandable GNN explanation. In addition, subgraph-based explainers like SubgraphX need to handle a large searching space, which leads to efficiency issues when generating explanations for dense or large scale graphs. In contrast, the number of the extracted motifs can be constrained by well-designed motif extraction rules, which means that using motifs as explanations can significantly reduce the search space. Another limitation of SubgraphX is that it needs to pre-determine a maximum number of nodes for its searching space. As the number of nodes in graphs varies greatly, it is hard to set a proper number for searching subgraphs. A large number will tremendously increase the computational resources, while a small number can limit the power of the explainer. To address the limitations of subgraph-based explainers, we propose a novel method that explicitly select important motifs as an explanation for a given graph. Compared to explainers based on subgraphs, our method generates explanations with motifs, which are statistically important and more human-understandable.

## 2.2 MOTIF EXTRACTION

This section introduces domain-specific motif extraction rules.

**Domain knowledge.** When working with data from different domains, motifs are extracted based on specific domain knowledge. For example, in biological networks, feed-forward loop, bifan, single-

input, and multi-input motifs are popular motifs, which have shown to have different properties and functions Alon (2007); Mangan & Alon (2003); Gorochowski et al. (2018). For graphs or networks in the engineering domain, the three-node feedback loop Leite & Wang (2010) and four-node feedback loop motifs Piraveenan et al. (2013) are important in addition to the feed-forward loop and bifan motifs. Motifs have also been shown to be important in computational Chemistry Yu & Gao (2022). The structures of these motifs are illustrated in Appendix 3.

**Extraction methods.** For molecule datasets, we can use sophisticated decomposition methods like RECAP Lewell et al. (1998) and BRICS Degen et al. (2008) algorithms to extract motifs. For other datasets that do not have mature extraction methods like biological networks and social networks, inspired by related works on graph feature representation learning Yu & Gao (2022); Bouritsas et al. (2022), we propose a general extraction method that only considers cycles and edges as motifs, which can cover most popular network motifs.

In particular, given a graph, we first extract all cycles out of it. Then, all edges that are not inside the cycles are considered motifs. We consider combining cycles with more than two coincident nodes into a motif. Although this method cannot extract complex motifs like single-input and multi-input motifs, it can generate the most important motifs, such as ring structures in biochemical molecules and the feed-forward loop motif. By adopting this simple but general motif extraction method, we can explain a GNN model without any domain knowledge, making our explanation model more applicable. Need to be noted that, even though the motif extraction rule cannot extract single-input and multi-input motifs, these motifs can be implicitly identified by our attention layer. Experiments in the table 1 demonstrate it.

Our methods can be easily applied to other domains by changing the motif extraction rules accordingly.

**Computational graph.** We define the computational graph of a given graph based on different tasks. The computational graph includes all nodes and edges contributing to the prediction. Since most GNNs follow a neighborhood-aggregation scheme, the computational graph usually depends on the architecture of GNNs, such as the number of layers. In graph classification tasks, all nodes and edges contribute to the final prediction. Thus, a graph itself is its computational graph in graph classification tasks. For node classification tasks, a target node's computational graph is the $L$-hop subgraph centered on the target node, where $L$ is the number of GNN layers. Here, we only consider motifs in the computational graph since those outside it are irrelevant to the predictions.

**Motif extraction.** Given a graph $G$, we extract all motifs based on the motif extraction method. If a motif has been extracted from the graph, it is added to a motif list $\mathcal{M}$. After searching the whole graph, there may be edges not in any motif. We regard each of them as a one-edge motif and add them to the motif list to retain the integrity of the graph information. At last, we can obtain the motif list $\mathcal{M} = [m_1, m_2, \ldots, m_t]$ in $G$.

## 2.3 GNN Explanation for Graph Classification Tasks

---
**Algorithm 1** MotifExplainer for graph classification tasks

---
**Input:** a set of graphs $\mathcal{G}$, labels for graphs $Y = \{y_1, ..., y_i, ..., y_n\}$, a pre-trained GNN $\Psi(\cdot)$, a pre-trained classifier $\xi(\cdot)$, motif extraction rule $\mathcal{R}$
**Initialization:** initial a trainable weight matrix $\boldsymbol{W}$
**for** graph $G_i$ in $\mathcal{G}$ **do**
    Graph embedding $\boldsymbol{j} = \Psi(G_i)$
    Create motif list $\mathcal{M} = \{m_1, ..., m_j, ..., m_t\}$ based on extraction rule $\mathcal{R}$
    Generate motif embedding for each motif $\boldsymbol{m}_j = \Psi(m_j)$
    Obtain an output score for each motif $s_j = \boldsymbol{m}_j \cdot \boldsymbol{W} \cdot \boldsymbol{h}$
    Train an attention weight for each motif $\alpha_j = \frac{\exp(s_j)}{\sum_{k=1}^{t} \exp(s_k)}$
    Acquire an alternative graph embedding $\boldsymbol{h}' = \sum_{k=1}^{t} \alpha_k \boldsymbol{m}_k$
    Output a prediction for the alternative graph embedding $y_i' = \xi(\boldsymbol{h}')$
    Calculate loss based on $y_i$ and $y_i'$ : loss $= f(y, y')$
    Update weight $\boldsymbol{W}$.
**end for**

---

## 2.4 Motif Embedding

After extracting motifs $\mathcal{M}$ from a given graph, we encode the feature representations for each motif. Given a pre-trained GNN model, we split it into two parts: a feature extractor $\Psi(\cdot)$ and a classifier $\xi(\cdot)$. The feature extractor $\Psi(\cdot)$ generates an embedding for the prediction target. In particular, $\Psi(\cdot)$ outputs graph embeddings in graph classification tasks, and outputs node embeddings in node classification tasks. The motif embedding is obtained in a graph classification task by feeding all motif node embeddings into a readout function. While in a node classification task, motif embedding encodes the influence of the motif on the node embedding of the target node. Thus, we feed the target node $k$ and a motif $m_j \in \mathcal{M}$ as a subgraph into the GNN feature extractor $\Psi(\cdot)$ and use the resulting target node embedding of $k$ as the embedding of the motif. To ensure the connectivity of the subgraph, we keep edges from the target node to the motif and mask features of irrelevant nodes.

This section introduces how to generate an explanation for a pre-trained GNN model in a graph classification task. We split the pre-trained GNN model into a feature extractor $\Psi(\cdot)$ and a classifier $\xi(\cdot)$. Given a graph $G$, its original graph embedding $\boldsymbol{h}$ is computed as $\boldsymbol{h} = \Psi(G)$. The prediction $y$ is computed by $y = \xi(\boldsymbol{h})$.

Based on the given graph, our method extracts a motif list from it and generates motif embedding $\boldsymbol{M} = [\boldsymbol{m}_1, \boldsymbol{m}_2, \ldots, \boldsymbol{m}_t]$ using the pre-trained feature extractor $\Psi(\cdot)$. Since the original graph embedding is directly related to the predictions, we identify the most important motifs by investigating relationships between the original graph embedding and motif embeddings. To this end, we employ an attention layer, which uses the original graph embedding $\boldsymbol{h} = \Psi(G)$ as query and motif embedding $\boldsymbol{M}$ as keys and values. The output of the attention layer is considered as a new graph embedding $\boldsymbol{h}'$. We interpret the attention scores as the strengths of relationships between the prediction and motifs. Thus, highly relevant motifs will contribute more to the new graph embedding. By feeding the new graph embedding $\boldsymbol{h}'$ into the pre-trained graph classifier $\xi(\cdot)$, a new prediction $y' = \xi(\boldsymbol{h}')$ is obtained. The loss based on $y$ and $y'$ evaluates the contribution of selected motifs to the final prediction, which trains the attention layer such that important motifs are selected to produce similar predictions to the original graph embedding. Formally, this explanation process can be represented as

$$\boldsymbol{h} = \Psi(G), y = \xi(\boldsymbol{h}), \tag{1}$$

$$M = [m_1, m_2, \ldots, m_t] = \text{MotifExtractor}(G), \tag{2}$$

$$\boldsymbol{M} = [\boldsymbol{m}_1, \boldsymbol{m}_2, \ldots, \boldsymbol{m}_t] = [\Psi(m_i)]_{i=1}^t, \tag{3}$$

$$\boldsymbol{h}' = \text{Attn}(\boldsymbol{h}, \boldsymbol{M}, \boldsymbol{M}), \tag{4}$$

$$y' = \xi(\boldsymbol{h}'), \tag{5}$$

$$\text{loss} = f(y, y'), \tag{6}$$

where $\text{Attn}(\cdot)$ is an attention layer and $f$ is a loss function. After training, we use the attention scores to identify important motifs. To our knowledge, our work first attempts to use the attention mechanism for GNN explanation. We want to mention that the attention mechanism is only a tool for selecting important motifs. Any other methods that can identify relevances between two feature vectors can be applied in our model. In addition, attention scores are only used in training, while we have other metrics for evaluation.

During testing, we use a threshold $\sigma/t$ to select important motifs, where $\sigma$ is a hyper-parameter and $t$ is the number of motifs extracted. The explanation includes the motifs whose attention scores are larger than the threshold. Algorithm 1 describes our GNN explanation method on graph classification tasks. In addition, we provide an illustration of the proposed MotifExplainer in Figure 1.

## 2.5 GNN Explanation for Node Classification Tasks

This section introduces how to generate an explanation for a node classification task. Given a graph $G$ and a target node $v_i$, we first construct a computational graph for $v_i$, which is an $L$-hop subgraph as described in Section 2.2. Then we extract motifs from the computational graph and generate motif embedding for each motif using the feature extractor $\Psi(\cdot)$. To keep the connectivity between a target node and a motif, we keep the shortest path between each node in the motif and the target node in an explanation graph. To reduce the impact of nodes on the path, we set irrelevant nodes' features to zero. After that, the proposed MotifExplainer employs an attention layer to identify important

---

**Algorithm 2** MotifExplainer for node classification tasks

---

**Input:** a graph $G$, labels for all nodes in the graph $Y = \{y_1, ..., y_i, ..., y_n\}$, a pre-trained GNN $\Psi(\cdot)$, a pre-trained classifier $\xi(\cdot)$, motif extraction rule $\mathcal{R}$
**Initialization:** initial a trainable weight matrix $W$, calculate all node embedding $\mathcal{H} = \{h_1, ..., h_i, ..., h_n\}$
**for** node $v_i$ in the graph $G$ **do**
    Original node embedding $h_i \in \mathcal{H}$
    Create motif list $\mathcal{M} = \{m_1, ..., m_j, ..., m_t\}$ based on extraction rule $\mathcal{R}$
    For each motif $m_j$, we keep the motif, the target node $v_i$ and the edges between them. Then we put this subgraph into the pre-trained GNN $\Psi(\cdot)$ and get a new node embedding of target node $v_i$ as the motif embedding $m_j$
    Obtain an output score for each motif $s_j = m_j \cdot W \cdot h_i$
    Train an attention weight for each motif $\alpha_j = \frac{\exp(s_j)}{\sum_{k=1}^{t} \exp(s_k)}$
    Acquire an alternative graph embedding $h_i' = \sum_{k=1}^{t} \alpha_k m_k$
    Output a prediction for the alternative graph embedding $y_i' = \xi(h_i')$
    Calculate loss based on $y_i'$ and $y_i$
    Update weight $W$ using back-propagation.
**end for**

---

motifs. The attention layer for node classification tasks is similar to the one for graph classification tasks, except that the query is the embedding of the target node. A node embedding is generated by feeding the whole graph into the feature extractor $\Psi(\cdot)$. The target node's output feature vector $h_i$ is used as the query vector in the attention layer, which outputs the new node embedding $h_i'$. Similarly, the new prediction $y' = \xi(h_i')$ is obtained by feeding $h_i'$ into the pre-trained classifier.

We use a threshold $\sigma/t$ during testing to identify important motifs as an explanation. Algorithm 2 describes the details of the MotifExplainer on node classification tasks. Formally, the different parts from Section 2.3 are represented as

$$h = \Psi(G)_i, y = \xi(h), \tag{7}$$
$$G_c = \text{ComputationGraph}(G, v_i), \tag{8}$$
$$M = [m_1, m_2, \ldots, m_t] = \text{MotifExtractor}(G_c). \tag{9}$$

Then, Eq. (3 - 6) are applied to compute loss for training the attention layer.

## 3 EXPERIMENTAL STUDIES

We conduct experiments to evaluate the proposed methods on both real-world and synthetic datasets.

### 3.1 DATASETS AND EXPERIMENTAL SETTINGS

We evaluate the proposed methods using different downstream tasks on seven datasets (MUTAG, PTC, NCI1, PROTEINS, IMDB, 2Motifs, Shape) to demonstrate the effectiveness of our model.The details of datasets and experimental settings are introduced in Appendix D.

**Baselines.** We compare our MotifExplainer model with several state-of-the-art baselines: GNNExplainer, SubgraphX, PGExplainer, and ReFine. We also build a model that uses the same attention layer as MotifExplainer but assigns weights to edges instead of motifs. Noted that all methods are compared in a fair setting. During prediction, we use $\sigma = 1$ to control the size of selected motifs. Unlike other methods, we do not explicitly set a fixed number for selected edges as explanations, enabling maximum flexibility and capability when selecting important motifs.

**Evaluation metrics.** A fundamental criterion for explanations is that they must be human-explainable, which means the generated explanations should be easy to understand. Taking the BA-2Motif as an example, a graph label is determined by the house structure attached to a base BA graph. A good explanation of GNNs on this dataset should highlight the house structure. To this end, we perform qualitative analysis to evaluate the proposed method.

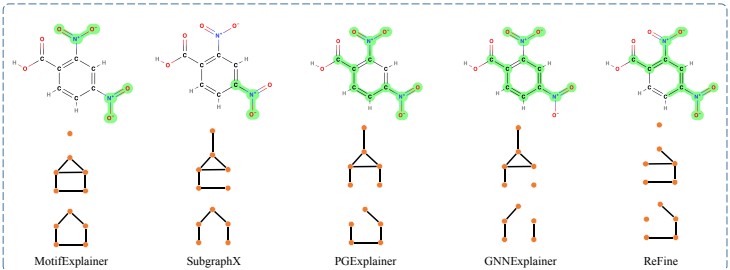

Figure 2: Visualization of explanation results from different explanation models on three datasets. The generated explanations are highlighted by green and bold edges. Three rows are results on the MUTAG dataset, the BA-Shape dataset, and the BA-2Motif dataset, respectively. We only show the motif-related edges for two synthetic datasets to save space.

Even though qualitative analysis/visualizations can provide insight into whether an explanation is reasonable for human beings, this assessment is not entirely dependable due to the lack of ground truth in real-world datasets. Thus, we employ three quantitative evaluation metrics to evaluate our explanation methods. We use the **Accuracy** metric to evaluate models for synthesis datasets with ground truth. Here, we use the same settings as GNNExplainer and PGExplainer. In particular, we regard edges inside ground truth motifs as positive edges and edges outside motifs as negative.

An explainer aims to answer a question that when a trained GNN predicts an input, which part of the input makes the greatest contribution. To this end, the explanation selected by an explainer must be unique and discriminative. Intuitively, the explanation obtained by the explainer should obtain similar prediction results as the original graph. Also, the explanation is in a reasonable size. Thus, following Yuan et al. (2020b), we use **Fidelity** and **Sparsity** metrics to evaluate the proposed method on real-world datasets. In particular, the Fidelity metric studies the prediction change by keeping important input features and removing unimportant features. The Sparsity metric measures the proportion of edges selected by explanation methods. Formally, they are computed by

$$\text{Fidelity} = \frac{1}{N} \sum_{i=1}^{N} \left( \Psi(G_i)_{y_i} - \Psi(G_i^{p_i})_{y_i} \right), \tag{10}$$

$$\text{Sparsity} = \frac{1}{N} \sum_{i=1}^{N} \left( 1 - \frac{|p_i|}{|G_i|} \right), \tag{11}$$

where $p_i$ is an explanation for an input graph $G_i$. $|p_i|$ and $|G_i|$ denote the number of edges in the explanation, and the number in the original input graph, respectively.

## 3.2 QUALITATIVE RESULTS

In this section, we visually compare the explanations of our model with those of state-of-the-art explainers. Some results are illustrated in Figure 2, with generated explanations highlighted. We report the visualization results of the MUTAG dataset in the first row. Unlike BA-Shape and BA-2Motif, MUTAG is a real-world dataset and does not have ground truth for explanations. We need to leverage domain knowledge to analyze the generated explanations. In particular, carbon rings with chemical groups $NH_2$ or $NO_2$ tend to be mutagenic. As mentioned by PGExplainer, carbon rings appear in both mutagen and non-mutagenic graphs. Thus, the chemical groups $NH_2$ and $NO_2$ are more important and considered as the ground truth for explanations. From the results, our MotifExplainer can accurately identify $NH_2$ and $NO_2$ in a graph while other models can not. PGExplainer identifies some extra unimportant edges. SubgraphX produces subgraphs as explanations that are neither motifs nor human-understandable. Our proposed GNN explainer can consider motif information and generate better explanations on molecular graphs. Note that neither $NH_2$ nor $NO_2$ is explicitly included in our motif extraction rules. The explanation is generated by identifying bonds in these groups, which means that our method can be used to find motifs.

We show the visualization results of the BA-Shape dataset in the second row of Figure 2. In this dataset, a node's label depends on its location as described in Section 3.1. Thus, an explanation generated by an explainer for a target node should be the motif. We consider the selected edges on

the motif to be positive and those not on the motif negative. From the results, our MotifExplainer can accurately mark the motif as the explanation. However, other models select a part of the motif or include extra non-motif edges. The third row of Figure 2 shows the visualization results on the BA-2Motif dataset, which is also a synthetic dataset. From Section 3.1, a graph's label is determined by the motif attached to the base graph: the five nodes house-like motif or the five nodes cycle motif. Thus, we treat all edges in these two motifs to be positive and the rest of edges to be negative. From the results, we can see that our MotifExplainer can precisely identify both the house-like motif and the cycle motif in a graph without including non-motif edges. In contrast, other models select edges far from the motif. More qualitative analysis results on MUTAG dataset are reported in Appendix E.

Table 1: Results on quantitative studies for different explanation methods. Note that since the Sparsity cannot be fully controlled, we report Fidelity scores (The less the better) under similar Sparsity levels for five real-world datasets. For two synthetic datasets, BA-Shape and BA-2Motif, we report accuracy. $S$ is the sparsity value. $K$ is the maximum number of edges required by baseline models. Our MotifExplainer does not need this required hyper-parameter. The best performances on each dataset are shown in **bold**.

| | MUTAG $S$=0.7 | PTC $S$=0.7 | NCI1 $S$=0.7 (Fidelity) | PROTEINS $S$=0.7 | IMDB $S$=0.7 | 2Motifs $K$=5 | Shape $K$=5 (Accuracy) |
|---|---|---|---|---|---|---|---|
| GNNExplainer | 0.260 | 0.441 | 0.365 | 0.453 | 0.365 | 0.742 | 0.925 |
| PGExplainer | 0.241 | 0.388 | 0.402 | 0.521 | 0.225 | 0.926 | 0.963 |
| SubgraphX | 0.287 | 0.227 | 0.303 | 0.021 | 0.167 | 0.774 | 0.874 |
| ReFine | 0.221 | 0.349 | 0.409 | 0.435 | 0.127 | 0.932 | 0.954 |
| **MotifExplainer** | **0.031** | **0.129** | **0.115** | **-0.030** | **0.101** | **1.0** | **1.0** |

## 3.3 QUANTITATIVE RESULTS

Table 2: Quantitative results on PTC and NCI1 dataset. The evaluation metric is Fidelity (The less the better). $S$ is the sparsity value. The best performances on each dataset are shown in **bold**.

| | PTC (Fidelity) | | | NCI (Fidelity) | | |
|---|---|---|---|---|---|---|
| | $S$=0.6 | $S$=0.7 | $S$=0.8 | $S$=0.6 | $S$=0.7 | $S$=0.8 |
| GNNExplainer | 0.3835 | 0.4406 | 0.4947 | 0.3612 | 0.3653 | 0.3648 |
| PGExplainer | 0.3653 | 0.3886 | 0.3917 | 0.4013 | 0.4029 | 0.4045 |
| ReFine | 0.3268 | 0.3499 | 0.3575 | 0.4028 | 0.4093 | 0.4115 |
| SubgraphX | 0.2062 | 0.2274 | 0.2643 | 0.1697 | 0.3036 | 0.4075 |
| **MotifExplainer** | **0.1162** | **0.1299** | **0.2256** | **0.1002** | **0.1154** | **0.1297** |

Table 3: Quantitative results on PROTEINS and IMDB-B dataset. The evaluation metric is Fidelity (The less the better). $S$ is the sparsity value. The best performances on each dataset are shown in **bold**.

| | PROTEINS (Fidelity) | | | IMDB-B (Fidelity) | | |
|---|---|---|---|---|---|---|
| | $S$=0.6 | $S$=0.7 | $S$=0.8 | $S$=0.6 | $S$=0.7 | $S$=0.8 |
| GNNExplainer | 0.4558 | 0.4535 | 0.4947 | 0.1577 | 0.3653 | 0.3098 |
| PGExplainer | 0.5215 | 0.5214 | 0.5207 | 0.1801 | 0.2253 | 0.2784 |
| ReFine | 0.3399 | 0.4354 | 0.4974 | 0.0952 | 0.1278 | 0.1829 |
| SubgraphX | 0.0138 | 0.0211 | 0.0398 | 0.1342 | 0.1671 | 0.1955 |
| **MotifExplainer** | **-0.0140** | **-0.0300** | **-0.0558** | **0.0757** | **0.1011** | **0.1125** |

Under inductive learning settings, we compared our methods with other state-of-the-art models on graph classification tasks with MUTAG, PTC, NCI1, PROTEINS, IMDB-BINARY, and BA-2Motifs datasets. Under transductive learning settings, we compare our proposed method with other state-

of-the-art models in terms of node classification accuracy. We report node classification accuracies on datasets BA-Shape.

To fully demonstrate the importance of motifs in GNN explanation, we build an explanation model named AttnExplainer, which directly uses an attention model to score and select edges instead of motifs. In AttnExplainer, an edge embedding is generated by taking the mean of its two ending nodes embedding. We will discuss more on AttnExplainer in section 3.4.

Table 4: Quantitative results on MUTAG dataset. The evaluation metric is Fidelity (The less the better). $S$ is the sparsity value. The best performances on each dataset are shown in **bold**.

| | **MUTAG** (Fidelity) | | | | |
| | $S$=0.4 | $S$=0.5 | $S$=0.6 | $S$=0.7 | $S$=0.8 |
|---|---|---|---|---|---|
| GNNExplainer | 0.153 | 0.184 | 0.219 | 0.260 | 0.307 |
| PGExplainer | 0.133 | 0.154 | 0.194 | 0.241 | 0.297 |
| SubgraphX | 0.214 | 0.233 | 0.254 | 0.287 | 0.376 |
| ReFine | 0.075 | 0.124 | 0.180 | 0.221 | 0.311 |
| **AttnExplainer** | 0.085 | 0.111 | 0.133 | 0.166 | 0.182 |
| **MotifExplainer** | **0.025** | **0.053** | **0.054** | **0.031** | **0.028** |

We first report the Fidelity scores under the same Sparsity value on five real-world datasets and the accuracy on the other two synthetic datasets. The results are summarized in Table 1. From the results, our MotifExplainer consistently outperforms previous state-of-the-art models on all seven datasets under a Sparsity value equal to 0.7. Note that our method achieves 100% accuracy on two synthetic datasets and at least 2.6% to 19.0% improvements on the real-world datasets.

In addition, more Fidelity scores on the real-world datasets are shown in Table 4, 2, 3. Table 4 compares our method with other baselines on the MUTAG dataset under different Sparsity values from 0.4 to 0.8. We can see that our method achieves the best performance in terms of Fidelity and Sparsity on the evaluated dataset. Table 2 and 3 show the performance of our model on four real-world datasets. We notice that MotifExplainer surpasses all the baselines by a notable margin.

Our model can maintain good performances when Sparsity is high. In particular, in the case of high Sparsity, the explanation contains a very limited number of edges, which shows that our model can identify the most important structures for GNN explanations. Using motifs as basic explanation units, our model can preserve the characteristics of motifs and the connectivity of edges which provide a robust explainer compare to other baselines.

To this end, MotifExplainer can learn to discover the motif-based explanation in a global view of the whole dataset and thus outperforms all baselines on all seven datasets, which demonstrates the effectiveness of our MotifExplainer.

## 3.4 ABLATION STUDIES

Our MotifExplainer employs an attention model to score and select the most relevant motifs to explain a given graph. To demonstrate the effectiveness of using motifs as basic explanation units, we build a new model named AttnExplainer that uses edges as basic explanation units and apply an attention model to select relevant edges as explanations. We compare our MotifExplainer with AttnExplainer on three datasets: BA-Shape, BA-2Motif, MUTAG. The results are summarized in Table 5, and Table 4. From the results, our model can consistently outperform AttnExplainer. This is because motifs can better obtain structural information than edges by using motifs as the basic unit for the explanation.

Table 5: Results for AttnExplainer and MotifExplainer on three datasets. Sparsity $S = 0.7$ for the MUTAG dataset, and $K = 5$ for two synthetic datasets.

| | **MUTAG** | **2Motif** | **Shape** |
|---|---|---|---|
| AttnExplainer | 0.166 | 0.934 | 0.955 |
| MotifExplainer | **0.031** | **1.0** | **1.0** |

## 3.5 ATTENTION LAYER STUDIES

In this section, we investigate the capability of the attention layer in learning new graph embeddings from mo-

Table 6: The accuracy of reconstructions on five real-world datasets.

| | **MUTAG** | **NCI1** | **PROTEINS** | **PTC** | **IMDB** |
|---|---|---|---|---|---|
| Accuracy | 99.98 | 99.97 | 98.65 | 99.34 | 98.46 |

tif embeddings to accurately represent the original graph structure. Table 6 presents the accuracy

with which our attention layer reconstructs the original graph embedding using these motif embeddings. The results demonstrate that our method achieves a very high accuracy, indicating that the learned graph embeddings effectively approximate the behavior of the original graph embeddings. This high level of accuracy suggests that the attention layer is successfully capturing and integrating the essential interaction of motifs, thereby validating the effectiveness of attention weights in explanation tasks. Due to the page limits, we have more threshold studies, efficiency studies, and motif studies in Appendix.

## 4 RELATED WORK

The research on GNN explainability is mainly divided into two categories: instance-level explanation and model-level explanation. Instance-level GNN explanation can also be divided into four directions, namely gradients/features-based methods, surrogate methods, decomposition methods, and perturbation-based methods. Gradients/features-based methods use gradients or hidden feature map values as the approximations of an importance score of an input. Recently, several methods have been employed to explain GNNs like SA Baldassarre & Azizpour (2019), Guided-BP Baldassarre & Azizpour (2019), CAM Pope et al. (2019), Grad-CAM Pope et al. (2019). The main difference between these methods is the process of gradient back-propagation and how different hidden feature maps are combined. The basic idea of surrogate methods is using a simple and explainable surrogate model to approximate the predictions of GNNs. Several methods have been introduced recently, such as GraphLime Huang et al. (2020), RelEx Zhang et al. (2021), and PGM-Explainer Vu & Thai (2020). Decomposition methods like LRP Baldassarre & Azizpour (2019), Excitation BP Pope et al. (2019), GNN-LRP Schnake et al. (2020) and DEGREE Feng et al. (2021) measure the importance of input features by decomposing original predictions into several terms. The last method is the perturbation-based method. Along this direction, GNNExplainer Ying et al. (2019) learns soft masks for edges and node features to generate an explanation via mask optimization. PGExplainer Luo et al. (2020) learns approximated discrete masks for edges by using domain knowledge. GraphMask Schlichtkrull et al. (2020) proposes a method for explaining the edge importance by generating an edge mask for each GNN layer. ZORRO Funke et al. (2020) studies discrete masks to select significant input nodes and node features. Causal Screening Wang et al. (2020) employs the causal attribution of different edges in an input graph as an explanation. SubgraphX Yuan et al. (2021) employs the Monte Carlo Tree Search algorithm to search possible subgraphs and uses the Shapley value to measure the importance of subgraphs and choose a subgraph as the explanation. ReFine Wang et al. (2021) proposes an idea of generating multi-grained explanations. There are also some reinforcement learning based explainers Shan et al. (2021); Wang et al. (2022). Model-level explanation methods in the field are currently under-researched, with only a few studies addressing this issue. These methods can be divided into two categories. The first is concept-based methods. GCExplainer Magister et al. (2021) incorporates concepts into GNN explanations. PAGE Shin et al. (2022) discovers propotypes as explanations. GLGExplainer Azzolin et al. (2022) adopts prototype learning to identify data prototypes. GCNeuron Xuanyuan et al. (2023), employs human-defined rule in natural language and use compositional concepts with the highest scores for global explanations. The second category is generation-based methods. XGNN Yuan et al. (2020a) uses deep reinforcement learning to generate explanation graphs node by node. GNNInterpreter Wang & Shen (2022), alternatively, learns a probabilistic model that identifies the most discriminative graph patterns for explanations.

## 5 CONCLUSION

This work proposes a novel model-agnostic motif-based GNN explainer to explain GNNs by identifying important motifs, which are recurrent and statistically significant patterns in graphs. Our proposed motif-based methods can provide better human-understandable explanations than methods based on nodes, edges, and regular subgraphs. Given a graph, We first extract motifs from a graph using motif extraction rules based on domain knowledge. Then, motif embedding for each motif is generated using the feature extractor from a pre-trained GNN. After that, we train an attention model to select the most relevant motifs based on attention weights and use these selected motifs as an explanation for the input graph. Experimental results show that our MotifExplainer can significantly improve explanation performances from quantitative and qualitative aspects.

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

# A   POPULAR MOTIFS IN NETWORK SCIENCE

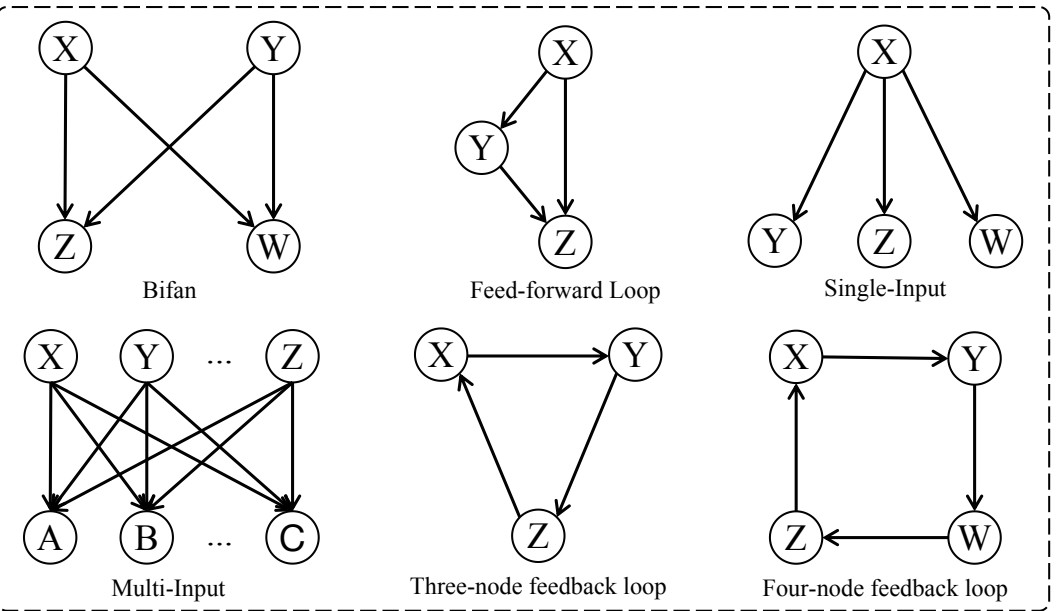

Figure 3: Popular motifs in biological and engineering networks.

# B   STATISTICS AND PROPERTIES OF DATASETS

Table 7: Statistics and properties of seven datasets.

|  | MUTAG | PTC | NCI1 | PROTEINS | IMDB | BA-2Motif | BA-Shape |
|---|---|---|---|---|---|---|---|
| # Edges (avg) | 30.77 | 14.69 | 32.30 | 72.82 | 96.53 | 25.48 | 4110 |
| # Nodes (avg) | 30.32 | 14.29 | 29.87 | 39.06 | 19.77 | 25.0 | 700 |
| # Graphs | 4337 | 344 | 4110 | 1113 | 1000 | 1000 | 1 |
| # Classes | 2 | 2 | 2 | 2 | 2 | 2 | 4 |

# C   DATASETS

MUTAG Kazius et al. (2005); Riesen & Bunke (2008) is a chemical compound dataset containing 4,337 molecule graphs. Each graph can be categorized into mutagen and non-mutagen which indicates the mutagenic effects on Gramnegative bacterium Salmonella typhimurium.

PTC Kriege & Mutzel (2012) is a collection of 344 chemical compounds reporting the carcinogenicity for rats.

NCI1 Wale et al. (2008) is a balanced subset of datasets of chemical compounds screened for activity against non-small cell lung cancer and ovarian cancer cell lines respectively.

PROTEINS Dobson & Doig (2003) is a protein dataset classified as enzymatic or non-enzymatic. The nodes represent amino acids, and if the distance between the two nodes is less than 6 Angstroms, the two nodes are connected by an edge.

IMDB-BINARY Yanardag & Vishwanathan (2015) is a movie collaboration dataset that consists of the ego-networks of 1,000 actors/actresses who played roles in movies in IMDB. In each graph, nodes represent actors/actress, and there is an edge between them if they appear in the same movie

BA-2Motifs Luo et al. (2020) is a synthetic graph classification dataset. It contains 800 graphs, and each graph is generated from a Barabasi-Albert (BA) base graph. Half graphs are connected with house-like motifs, while the rest are assigned with five-node cycle motifs. The labels of graphs are assigned based on the associated motifs. All node features are initialized as vectors with all 1s.

BA-Shapes Ying et al. (2019) is a synthetic node classification dataset. It contains a single base BA graph with 300 nodes. Some nodes are randomly attached with 80 five-node house structure motifs. Each node label is assigned based on its position and structure. In particular, labels of nodes in the base BA graph are assigned 0. Nodes located at the top/middle/bottom of the house-like network motifs are labeled with 1, 2, and 3, respectively. Node features are not available in the dataset.

## D    EXPERIMENTAL SETTINGS

For the pre-trained GNN, we use a 3-layer GCN as a feature extractor and a 2-layer MLP as a classifier on all datasets. The GCN model is pre-trained to achieve reasonable performances on all datasets. We use Adam optimizer for training. We set the learning rate to 0.01.

Real World Datasets: We employ a 3-layer GCNs to train all five real world datasets. The input feature dimension is 7 and the output dimensions of different GCN layers are set to 64, 64, 64, respectively. We employ mean-pooling as the readout function and ReLU as the activation function. The model is trained for 170 epochs with a learning rate of 0.01. We study the explanations for the graphs with correct predictions.

BA-Shape: We use a 3-layer GCNs and an MLP as a classifier to train the BA-Shape dataset. The hidden dimensions of different GCN layers are set to 64, 64, 64, respectively. We employ ReLU as the activation function. The model is trained for 300 epochs with a learning rate of 0.01. The validation accuracy of the pre-trained model can achieve 100%. We study the explanations for the whole dataset.

BA-2Motifs: We use a 3-layer GCNs and an MLP as a classifier to train the BA-2Motif dataset. The hidden dimensions of different GCN layers are set to 64, 64, 64, respectively. We employ mean-pooling as the readout function and ReLU as the activation function. The model is trained for 300 epochs with a learning rate of 0.01. The validation accuracy of the pre-trained model can be 100%, which means the model can perfectly generate the distribution of the dataset. We study the explanations for the whole dataset.

We conduct experiments using one Nvidia 2080Ti GPU on an AMD Ryzen 7 3800X 8-Core CPU. Our implementation environment is based on Python 3.9.7, Pytorch 1.10.1, CUDA 10.2, and Pytorch-geometric 2.0.3.

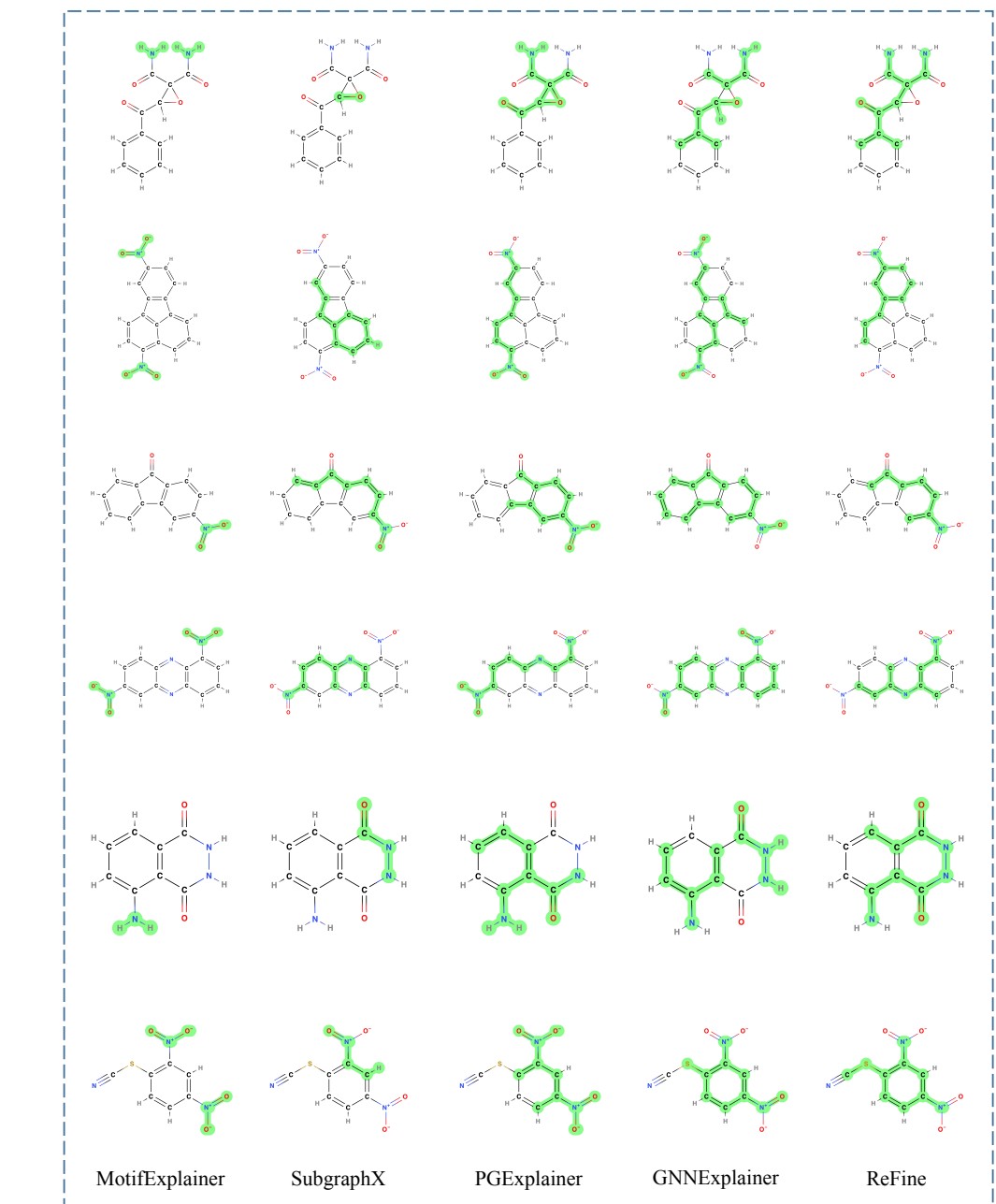

Figure 4: Visualization of explanation on MUTAG dataset.

# E ADDITIONAL QUALITATIVE RESULTS

# F    THRESHOLD STUDIES

Table 8: The study of the threshold on MUTAG dataset.

| Threshold $\sigma$ | 1.0 | 1.2 | 1.5 | 1.7 | 2.0 |
|---|---|---|---|---|---|
| **Sparsity** | 0.4 | 0.5 | 0.6 | 0.7 | 0.8 |
| **Fidelity** | 0.025 | 0.053 | 0.054 | 0.031 | 0.028 |

Our MotifExplainer uses a threshold $\sigma$ to select important motifs as explanations during inference. Since $\sigma$ is an important hyper-parameter, we conduct experiments to study its impact using Sparsity and Fidelity metrics. The performances of MotifExplainer using different $\sigma$ values on the MUTAG dataset are summarized in Table 8. Here, we vary the $\sigma$ value from 1.0 to 2.0 to cover a reasonable range. We can observe that when the threshold is larger, the Sparsity of explanations increases, and the performances in terms of Fidelity gradually decrease. This is expected since fewer motifs selected will be selected when the threshold becomes larger. Thus, the size of explanations becomes smaller, and the Sparsity value becomes larger. Note that even when the Sparsity reaches a high value of 0.8, our model can still perform well. This shows that our model can accurately select the most important motifs as explanations, demonstrating the advantage of using motifs as GNN explanations.

# G    EFFICIENCY STUDIES

Table 9: Results on efficiency studies on MUTAG dataset, Training contains both pre-processing/motif extraction step and model training time. Inference time is the average time consumed to obtain an explanation for a graph.

| Method | Inference | Training |
|---|---|---|
| GNNExplainer | 24.3s | 0s |
| PGExplainer | 0.03s | 740s |
| SubgraphX | 96.7s | 0s |
| ReFine | 0.83s | 946s |
| MotifExplainer | 0.02s | 363s |

We study the efficiency of our proposed model in terms of the training time and the inference time. For models that need to be trained, such as PGExplainer and ReFine, training and evaluation processes are separate. We report training and inference time separately. In our proposed method, the training time includes three parts: motif extraction, motif embedding construction, and the training of the attention model. For models that do not require training, their training time will be 0. For each model, we run it on the MUTAG dataset and show the averaging time consumed to obtain explanations for each graph. Table 9 shows the comparison results with four state-of-the-art GNN explanation models: MotifExplainer, SubgraphX, PGExplainer, GNNExplainer, and ReFine. From the results, our model has the shortest inference time among models. Compared to PGExplainer and ReFine, our model requires significantly less training time. From this point, the proposed method is efficient and feasible in real-world applications.

# H  MOTIF STUDIES

Table 10: The size of motif lists generated by our proposed motif extraction method on five real-world datasets.

|      | MUTAG | NCI1 | PROTEINS | PTC | IMDB |
|------|-------|------|----------|-----|------|
| **Size** | 148 | 296 | 1584 | 55 | 377 |

The quality and size of the motif list significantly influence the performance of our model. In this section, we study the efficiency of our proposed motif extraction method. Table 10 shows the size of motif lists extracted by our proposed extraction method on five real-world datasets. We can see that since we only consider cycles and edges as motifs, the size of the lists can be well controlled, which not only reduces the complexity of the model but also allows different graphs to share more motif information, resulting in better interpretation.

Table 11: The extraction time of motif lists generated by our proposed motif extraction method on five real-world datasets.

|      | MUTAG | NCI1 | PROTEINS | PTC | IMDB |
|------|-------|------|----------|-----|------|
| **Time(s)** | 103 | 90 | 19 | 0.9 | 33 |

We care about the time complexity of our motif extraction method. The complexity depends on the cycle basis generation algorithm. Currently, the most widely used cycle-basis extraction method is $O(n^3)$. We report the extraction time of motif lists on five real-world datasets in table 11. We can see that all five extraction procedures have been done in an acceptable time. Also, since we treat motif extraction as a pre-processing step, it will not affect the training/inference part of our method. We will additionally study efficiency in the next section.

