# OpenReview forum: "MotifExplainer: a Motif-based Graph Neural Network Explainer"
_ICLR.cc/2025/Conference — ICLR 2025 Conference Withdrawn Submission_

### Official Review · Reviewer_kHGc · 2024-10-17

**Soundness:** 2
**Presentation:** 1
**Contribution:** 2
**Rating:** 3
**Confidence:** 4

**Summary:**

This paper proposed a simple but effective method to explain GNNs at the instance-level. It first identifies motifs by domain knowledge, then  feeds each motif to the GNN to obtain the motif embedding. Finally, they build an attention-based network to obtain the attention weights of each motif in each graph instance, which are identified as the importance of the motifs.

**Strengths:**

1. Proposed method is simple but effective.
2. Good empirical results on Fidelity$-$ and Accuracy metrics compared with some old methods.

**Weaknesses:**

1. The baselines and related work are old. Recent works such as [r1,r2,r3,r4,r5] should be discussed and compared.

2. Presentation is very poor. Citation style needs to be corrected. \citep and \citet should be properly used. Some tables are confusing. See questions.

3. To extract motifs, domain knowledge are required. This make it impossible to be applied to a variety of real world tasks where the domain knowledge is unknown.

4. The fidelity evaluated in this paper is different from the one used in the paper of SubgraphX. Why not use their metric? We'd like to see how MotifExplainer performs on common metrics.

5. Feeding the motifs to GNNs, and training additional attention network will result in more computational cost. Can you also provide efficiency analysis?

[r1] Zhang, et al. Gstarx: Explaining graph neural networks with structure-aware cooperative games. Advances in Neural Information Processing Systems, 35:19810–19823, 2022.

[r2] Rong, et al. "Efficient gnn explanation via learning removal-based attribution." ACM Transactions on Knowledge Discovery from Data (2023).

[r3] Lu, et al. "GOAt: Explaining Graph Neural Networks via Graph Output Attribution." The Twelfth International Conference on Learning Representations, 2023.

[r4] Li, et al. DAG matters! GFlownets enhanced explainer for graph neural networks. In The Eleventh International Conference on Learning Representations, 2023.

[r5] Pereira, et al. Distill n’explain: explaining graph neural networks using simple surrogates. In International Conference on Artificial Intelligence and Statistics, pp. 6199–6214. PMLR, 2023.

**Questions:**

1. What is the metric for the results shown in Table 5? Why for MUTAG, the smaller the better, but for the other two the larger the better?

---

### Official Review · Reviewer_D26j · 2024-11-01

**Soundness:** 2
**Presentation:** 3
**Contribution:** 2
**Rating:** 3
**Confidence:** 4

**Summary:**

This paper proposes an explainable method for graph data by providing explanations using motifs, which represent subgraphs within the original graph that play a critical role in prediction. To generate explanations for a pretrained GNN model, they first extract motifs from the original graph using off-the-shelf extraction algorithms (e.g., BRICS, RECAP) or a proposed extraction method that generalizes by only considering cycles and edges as motifs. They then determine the importance of each motif by training an attention weight for each one. In experiments, they present both qualitative and quantitative results to demonstrate the superiority of their explanation method.

**Strengths:**

1. Considering motifs within the original graph makes the explanation more human-understandable.

2. The paper is well-written and easy to follow.

3. The method is intuitive and easy to understand.

**Weaknesses:**

1. The method appears highly dependent on the motif-extraction algorithm, which is not a contribution of this paper. For example, in the case of the MUTAG dataset, without domain knowledge, if the proposed motif extraction algorithm (a cycle-based extraction method) is used, $NH_2$ and $NO_2$ are unlikely to be identified as motifs that play a critical role in prediction. I strongly recommend that the authors show which motifs are extracted depending on the motif-extraction algorithm and compare the performance of the method accordingly.

2. PGIB [1], which has a closely related and similar motivation to this paper, should be included. PGIB also considers subgraphs (i.e., motifs) to provide explanations, sharing the same motivation of emphasizing the importance of motifs for explaining graph data. The paper should elaborate on its strengths compared to PGIB and include PGIB as a baseline in the experiments.

[1] NeurIPS'23, Interpretable Prototype-based Graph Information Bottleneck

**Questions:**

1. Performance differences depending on the motif-extraction algorithm need to be shown.

2. How does the proposed motif-extraction algorithm, which focuses on cycle structures, manage to extract $NO_2$ and $NH_2$ as motifs?

3. Compared to PGIB, the current SOTA method that shares a similar motivation (i.e., considering motifs) with this paper, what are the strengths of this paper?

4. The threshold $\sigma / t$ appears to have a significant effect on the final explanation; however, it also seems heuristic without guidance on how to determine it. How can we set this threshold when working with real-world datasets, and how can we evaluate whether the threshold is properly set?

5. Not all GNN prediction models may be explicitly divided into two parts: an embedder and a predictor. How can this method be applied in such cases?

---

### Official Review · Reviewer_Y5Xn · 2024-11-04

**Soundness:** 1
**Presentation:** 1
**Contribution:** 1
**Rating:** 3
**Confidence:** 4

**Summary:**

This paper introduces MotifExplainer, a novel method for explaining Graph Neural Networks (GNNs) by identifying important motifs within a graph.  MotifExplainer utilizes domain-specific motif extraction rules to identify these recurring substructures, creating motif embeddings through a pre-trained GNN’s feature extractor.
In graph classification, MotifExplainer aggregates motif embeddings to create a new graph embedding, while in node classification, it focuses on motifs that affect a specific node’s embedding. An attention layer highlights the most relevant motifs for predictions, aiming for more interpretable, human-understandable explanations. The approach is more efficient than subgraph-based methods by reducing the search space, and experiments show it provides high-quality explanations with improved interpretability and computational efficiency.

**Strengths:**

1. MotifExplainer focuses on statistically significant motifs rather than individual nodes or edges, providing more human-understandable explanations by highlighting recurring and functionally relevant substructures within graphs.

2. By reducing the search space to motifs rather than all possible subgraphs, MotifExplainer is computationally more efficient, making it suitable for dense or large-scale graphs.

**Weaknesses:**

1. The motif-based explanation approach is already a well-known method, with other papers[1,2,3,4] actively utilizing motifs for explainability. This paper needs to demonstrate its unique advantages and the necessity of its approach compared to these previous works.

- [1] Chen, Jialin, and Rex Ying. "Tempme: Towards the explainability of temporal graph neural networks via motif discovery." Advances in Neural Information Processing Systems 36 (2023): 29005-29028.
- [2] Ding, Feng, et al. "MEGA: Explaining Graph Neural Networks with Network Motifs." 2023 International Joint Conference on Neural Networks (IJCNN). IEEE, 2023.
- [3] Perotti, Alan, et al. "Graphshap: Motif-based explanations for black-box graph classifiers." arXiv preprint arXiv:2202.08815 (2022).
- [4] Zhang, Shichang, et al. "Motif-driven contrastive learning of graph representations." arXiv preprint arXiv:2012.12533 (2020).

2. In this model, cycles are used to extract motifs without domain knowledge. However, the paper needs to justify the validity of the statement "We consider combining cycles with more than two coincident nodes into a motif." Since motifs are central to this model, the model's validity hinges on how motifs are defined. The justification for the effectiveness of this approach in extracting motifs across various domains is insufficient.

3. The authors claim that their model addresses efficiency issues when generating explanations for dense or large-scale graphs. However, in Section G, they conducted experiments only on the simplest molecular dataset, the MUTAG dataset, without testing on large-scale data. To demonstrate the model's practical utility, efficiency experiments should also be performed on larger graph datasets, such as the IMDB dataset used by the authors, as well as on even larger datasets.

4. The model's performance heavily relies on motif extraction, which plays a critical role in explainability. It is necessary to show how performance varies with different motif extraction methods.

**Questions:**

The questions are listed in paper weakness.

---

### Official Review · Reviewer_pzr6 · 2024-11-04

**Soundness:** 2
**Presentation:** 3
**Contribution:** 3
**Rating:** 5
**Confidence:** 4

**Summary:**

This paper proposes a GNN explainer that uses motifs as the unit of explanation. By decomposing representations based on extracted motifs, it produces subgraph explanations. The proposed approach demonstrates its effectiveness across various datasets.

**Strengths:**

- Providing post-hoc explanations is crucial for training trustworthy GNNs.
 - Using motifs can be a valuable approach for interpretability, offering substantial potential impact.
 - The proposed method’s utility is supported through experiments on a range of datasets.

**Weaknesses:**

- It is unclear how this approach improves over existing subgraph-based explanation models, such as GLGExplainer [2].
 - The paper would benefit from comparisons with more recent XAI methods, such as D4Explainer [1] and MixupExplainer [3], along with subgraph-based explanation methods like GLGExplainer [2]. The most recent baseline in the experiment section of this paper was published in 2021.

References:

[1] Chen et al., "D4Explainer: In-distribution Explanations of Graph Neural Network via Discrete Denoising Diffusion," NeurIPS 2023.
[2] Azzolin, "Global Explainability of GNNs via Logic Combination of Learned Concepts," ICLR 2023.
[3] Zhang et al., "MixupExplainer: Generalizing Explanations for Graph Neural Networks with Data Augmentation," KDD 2023.

**Questions:**

- How are the most important motifs determined, and which motifs were defined and used as explanations in the experiments?
- In Algorithm 1, where does  h  originate?
- The proposed model includes motif extraction in the efficiency study, which is generally quite slow. How can it outperform existing models in speed? Could you also provide a time complexity analysis?

---

### Note · Authors · 2024-11-22

I have read and agree with the venue's withdrawal policy on behalf of myself and my co-authors.